# MoFE-Time: Mixture of Frequency Domain Experts for Time-Series Forecasting

## Abstract

Time series forecasting is a fundamental task with broad applications across various domains. Recently, inspired by the success of large language models (LLMs), foundation models for time series gained significant attention. However, most existing approaches directly adopt vanilla transformers, which underexplores the joint modeling of temporal and frequency characteristics, resulting in limited performance on complex time series. To address this, we propose MoFE-Time, a novel time series forecasting foundation model that integrates temporal and frequency-domain representations within a Mixture of Experts (MoE) framework. Specifically, we design Frequency and Time Cells (FTC) as experts following attention modules, and employ an MoE routing mechanism to construct multidimensional sparse representations of input signals. Extensive experiments on six public benchmarks demonstrate that MoFE-Time achieves new state-of-the-art results. Furthermore, we construct a proprietary real-world dataset, NEV-sales, to evaluate the model's practical effectiveness. MoFE-Time consistently outperforms competitive baselines on this dataset, demonstrating its potential for real-world commercial applications.

**Code** – `https://anonymous.4open.science/r/MoFE-Time-6E1D`

## 1 Introduction

Time series data serves as a fundamental modality within dynamic real-world systems Box et al. (2015); Zhang et al. (2024); Liang et al. (2024). As a critical task with significant demand across numerous fields—including sales, energy, climate science, quantitative finance, and urban computing Jin et al. (2023); Nie et al. (2024); Mao et al. (2024), the prediction of time series has been the subject of extensive research over the years. While progress has been made, modeling long-term dependencies, non-stationarity, and multi-scale periodicity in time series remains fundamentally challenging.

The ability of Large Language Models (LLMs) to model intricate patterns and dependencies in sequential data via attention mechanisms and large-scale pre-training naturally motivates their exploration for the prediction of time series. Promising initial efforts such as Time-MoE Shi et al. (2024), which builds a time series foundation model achieving significantly improved forecasting precision have aroused widespread concern. However, such approaches primarily focus on scaling existing architectures without explicit adaptations for the unique characteristics of time series data, thus ignoring the complex periodicity and non-stationarity inherent in time series modal.

Modeling the frequency-domain characteristics of time series data can effectively alleviate the above issues, facilitating the detection of latent periodic patterns. The standard practice employs the Fourier transform process to decompose the temporal sequence into frequency components. Nevertheless, current works Zhou et al. (2022); Wu et al. (2023b), merely convert input signals directly into the frequency domain for learning, thereby introducing additional mathematical constraints and lacking the ability to effectively learn signal frequency domain representations, which results in suboptimal performance and inadequate generalization.

A core challenge confronting current research can be summarized: how to develop a foundation model that effectively captures the distinctive characteristics of time series data to achieve superior cross-domain generalization. To address this challenge, we construct a time series forecasting network capable of directly learning the Fourier transform process that converts signals from the time domain to the frequency domain, thereby capturing the inherent properties of signals. We integrate frequency-domain learning with the Mixture-of-Experts (MoE) Liu et al. (2024a) architecture, as its intrinsic sparse properties naturally align with our objective, the ability to decompose distinct signals into separate spectral components for further analysis. In addition, we

posit that applying a pretraining-finetuning approach for time series models, can instill prior knowledge of signals into the model, thereby enhancing its predictive performance and generalization capabilities.

In general, we introduce MoFE-Time, a time series forecasting model based on the MoE architecture, which integrates both frequency and time features into each expert module to learn both the temporal features and frequency domain intrinsic properties of signals. By introducing reversible instance normalisation (RevIN) Kim et al. (2021) and time aggregation methods, the ability of the model to deal with non-stationarity and the applicability of the model to deal with variable-length series are improved. Frequency-Time Cell (FTC) for Domain Experts enhances the model's ability to capture the intrinsic properties of the signal's time and frequency domain, which contributes to improved predictive performance of the model. In experiments on six public benchmarks, MoFE-Time has achieved new **state-of-the-art** performance. Notably, the fine-tuned MoFE-Time can reduce MSE and MAE by **15.4%** and **7.1%** compared to the strongest baseline in our own constructed real-world commercial dataset.

The main contributions of our paper are:

- We propose MoFE-Time, a novel foundation model for time series forecasting, which introduces a Frequency-Time Cell (FTC) module to capture frequency-domain characteristics through Fourier transforms. This architecture specifically addresses the deficiency of existing foundation models in modeling frequency-domain information. By integrating the FTC module with MoE, our framework effectively enhances the model's capacity to learn and extrapolate temporal periodicity in time series.

- We build NEV-sales, a real-world dataset comprising daily customer count of about 500 sales centers of a leading new energy vehicle brand across one country from 2022 to 2025. Our model and dataset are open-sourced.

- The sufficient experimental results demonstrate that our method achieves state-of-the-art performance on six public datasets and our own constructed real-world dataset, confirming the effectiveness and applicability of MoFE-Time.

## 2 RELATED WORK

The advancement of pretrained models in natural language processing and computer vision has significantly improved the understanding of these modalities Dong et al. (2019); Selva et al. (2023). Recently, a shift towards general pretraining on extensive datasets for time series models has been observed. TimesFM Das et al. (2023), Moirai Liu et al. (2024b) demonstrated the effectiveness of leveraging large-scale pretraining for time series representation learning. Chronos Ansari et al. (2024) offers pre-trained models with up to 710 million parameters. Available in four sizes, these models effectively meet diverse application needs. Time-MoE Shi et al. (2024) marks a breakthrough with its scalable architecture using a sparse mixture of experts, boosting efficiency for large-scale forecasts. Using the Time-300B dataset, which includes 300 billion time points, it scales to 2.4 billion parameters.

As the field of temporal prediction continues to evolve, an increasing number of models are transitioning from learning time-domain signals to focusing on frequency-domain learning. FEDformer Zhou et al. (2022) integrates frequency-domain modules within transformer-based architectures to enhance temporal representation. TimesNet Wu et al. (2023b) further improves time series modeling by decomposing signals into multiple periods using the Fourier Transform. Despite these advances, the predominant paradigm remains the direct transformation of time series data into the frequency domain using techniques such as the Fourier Transform before model input. This strategy prevents the model from natively learning to transform and represent signals between temporal and spectral modalities. Moreover, it introduces risks such as spectral leakage Oppenheim (1999); Smith et al. (1997), which may degrade model performance.

## 3 METHOD

Given a historical input sequences $X^i = [x_1^i, x_2^i, \cdots, x_{T_x}^i]$ of fixed length $T_x$, we forcast the sequence values in a future window of specified length $T_y$, represented as $Y^i = [y_1^i, y_2^i, \cdots, y_{T_y}^i]$:

$$Y^i = \mathbf{f}(X^i; \theta), \tag{1}$$

where $\theta$ represents the time series model parameters.

We adopt the widely-regarded MoE architecture, and design expert networks utilizing Fourier analysis to capture intrinsic frequency domain characteristics as well as time domain characteristics of time series. An overview of our approach is given in Figure 1.

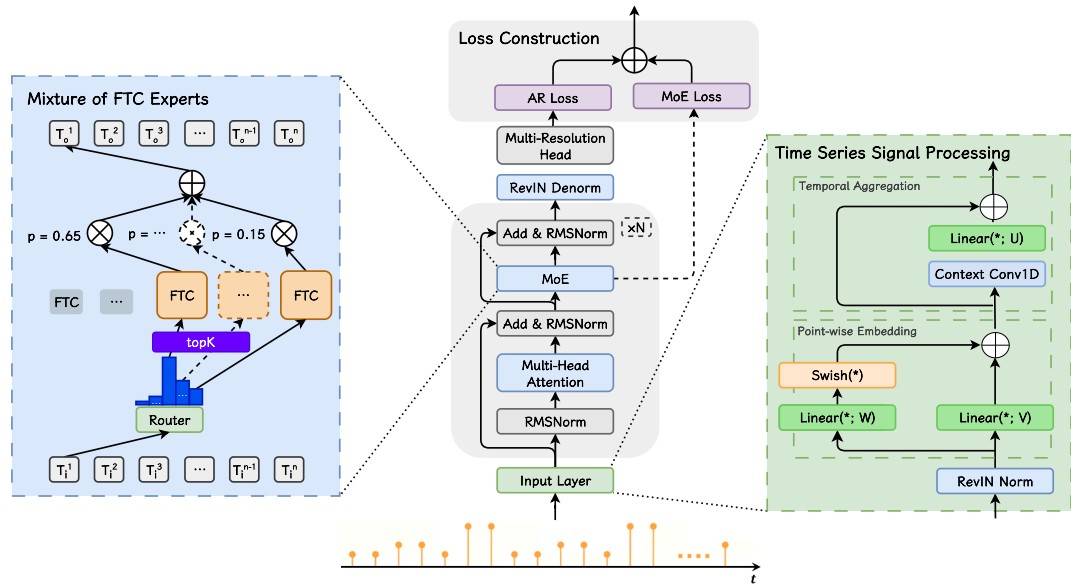

Figure 1: Overall architecture. Our network is based on the successful Time-MoE Shi et al. (2024) architecture. The proposed time series signal processing and Frequency-Time Cell (FTC) for Domain Experts are contained inside the blue and green boxes. By introducing RevIN and time aggregation methods, the ability of the model to deal with non-stationarity and the applicability of the model to deal with variable-length series are improved. FTC is enhancing the model's ability to capture the intrinsic properties of the signal's time and frequency domain, which contributes to improving the model's predictive performance.

## 3.1 TIME SERIES SIGNAL PROCESSING

We process the input time series data through a series of steps designed to address non-stationarity and capture both fine-grained and local temporal features.

Non-stationarity is an inherent property of real-world time series data, resulting in varying distributions even within short context windows of an individual time series. To effectively counteract the challenges posed by shifts in data distribution, we apply RevIN Kim et al. (2021) at the data input layer. RevIN performs instance-wise normalization and denormalization with learnable affine transformations, stabilizing the distribution of time series inputs while enhancing the model's ability to learn intrinsic stationary signal attributes in the frequency domain.

Next, we employ point-wise embedding for floating-point time series signals. Given the input sequence $X^i = [x_0^i, x_1^i, \cdots, x_{T_x}^i]$, where $x_j^i \in \mathbb{R}$, each value is projected independently into a $h$-dimensional latent space via two parallel linear transformations:

$$Z_j^W = W x_j^i, \quad Z_j^V = V x_j^i, \quad W, V \in \mathbb{R}^{h \times 1}. \tag{2}$$

To dynamically balance nonlinear transformations and linear projections, we introduce a learnable mixing coefficient $\alpha \in [0, 1]$, parameterized via a trainable scalar $\beta$. The gated output is formulated as:

$$E_j^i = \alpha \cdot \textbf{Swish}(Z_j^W) + (1 - \alpha) \cdot Z_j^V, \quad \alpha = \textbf{Sigmod}(\beta). \tag{3}$$

While the point-wise design ensures computational efficiency and expressivity, it lacks localized temporal context. To address this, we add a lightweight temporal aggregation module: dilated depthwise convolutions with kernel size $K = 3$, dilation rate $d = 2$ across the time axis $j$,

$$\hat{E}_j^i = E_j^i + \sum_{k=0}^{K-1} C_k \cdot E_{j-k \cdot d}^i, \quad C \in \mathbb{R}^{K \times h}, \tag{4}$$

$$E^i = \textbf{Linear}\left(\hat{E}^i\right) \in \mathbb{R}^{w \times h}, \tag{5}$$

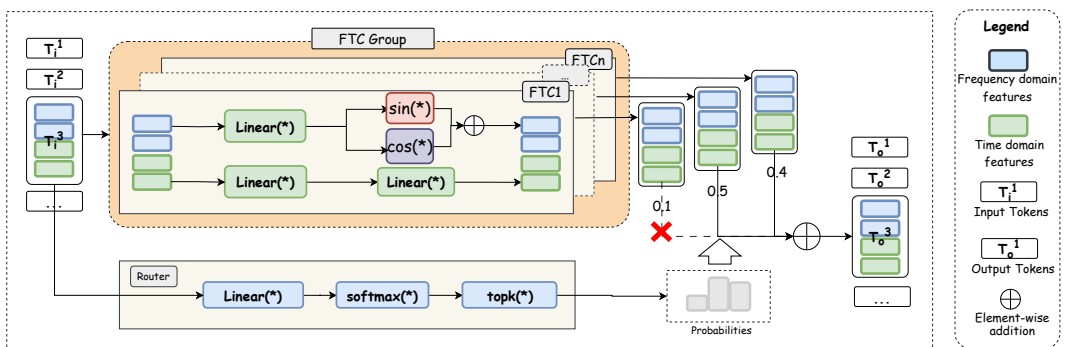

Figure 2: Details of the FTC module. We incorporate a Frequency-Time Cell (FTC) within the MoE framework as a specialized module to enable models to simultaneously capture both frequency-domain and time-domain characteristics of signals. By mapping a segment of the input signal through neural network layers and harmonic functions into a frequency-domain space, and subsequently integrating both time-domain and frequency-domain features before feeding them into the subsequent layers, we aim to enhance the model's capacity for representation and learning.

where $C$ denotes the convolution filters and the residual connection preserves the original embedding. The dilated convolution enlarges the receptive field for temporal context without significant parameter overhead.

## 3.2 FREQUENCY-TIME CELL FOR DOMAIN EXPERTS

We propose a Frequency-Time Cell (FTC) module within the MoE framework to enable models to simultaneously capture both frequency-domain and time-domain characteristics inherent in time series signals.

A time series within a context window can be regarded as a non-periodic discrete-time signal. Mathematically, such a signal can be decomposed into critical frequency components. Our network learns two key pieces of information:

1. Critical frequency components $\{\omega_i | i = 1, \cdots, k\}$,
2. Amplitudes of these critical frequency $\{a_i | i = 1, \cdots, k\}$.

The original time-domain signal can then be expressed as:

$$x_n = \sum_{i=1}^{k} \left( a_i e^{-j\omega_i n} \right). \tag{6}$$

In practice, the spectrum of a discrete non-periodic signal is inherently continuous, making explicit identification of important spectral components challenging. To address this, we utilize pretraining to allow the model to automatically discover $k \times h$ distinctive harmonic frequencies, where $k$ is the number of experts and $h$ is the number of distinct critical harmonic frequencies each expert holds.

For a given time point, the MoE routing algorithm selects the most significant $k \times h$ harmonics, subsequently performing a weighted sum by column, resulting in an $h$-dimensional vector representing the signal's periodicity. The routing weights correspond to the amplitudes of the different harmonics. Assume that the most critical frequencies learned by the $k$ chosen experts are:

$$F = [\Omega_1, \Omega_2, \cdots, \Omega_k], \tag{7}$$

where $\Omega_i = [\omega_1^i, \omega_2^i, \cdots, \omega_h^i]$ denotes the $h$ key frequencies learned by the $i$-th expert. Letting the routing weights be $A = [\alpha_1, \alpha_2, \cdots, \alpha_k]$, for the $n$-th token, the harmonic combinations are:

$$\underset{n=1,2,\cdots,h}{H[n]} = \sum_{i=1}^{k} \alpha_i e^{-j\omega_i n}. \tag{8}$$

The harmonic combination vector is concatenated with the time-domain signal vector to form a representation, serving as the output of the intermediate layer. The overall FTC structure is illustrated in Figure 2. Given the presence of positional encoding in the time-domain signal of each layer, inputs are initially divided into two

parts: one for frequency-domain modeling and another using a linear layer for temporal feature modeling. For frequency-domain modeling, segments $X_t$ are mapped to $X_f$:

$$X_f = \mathbf{Linear}(X_t; W), \tag{9}$$

then, via harmonic basis $e^{jwn}$, the $k$-th expert produces frequency-aware representations:

$$X_t^k = e^{jX_f^k}. \tag{10}$$

Since $X_t$ is a time-domain signal containing positional encoding, it effectively learns periodicity $\omega_i$, the design's foundation. The causal multi-head attention captures contextual dependencies, while the FTC—via MoE routing—selects and weights diverse frequency components, endowing the model with strong periodic pattern modeling capacity. To improve robustness, we use both sine and cosine harmonic bases, aggregating their outputs through direct summation in our implementation.

## 3.3 SUPERVISION

We supervise the time series predictions using two components:

**Autoregressive Loss.** Large-scale time series data often contain significant noise and outliers, which can adversely affect model training stability and generalization. To address this, we employ the Cauchy loss Mlotshwa et al. (2022) as our regression loss. Compared to conventional loss functions like MSE, the Cauchy loss better handles heavy-tailed noise in time series data, thus offering improved robustness and generalizability:

$$\mathcal{L}_{\mathrm{ar}}(y_t, \hat{y}_t) = \frac{c^2}{2} \log(1 + (\frac{y - \hat{y}}{c})^2), \tag{11}$$

where $c \in (0, \infty)$ is a hyperparameter controlling the robustness of the loss.

**MoE Auxiliary Loss.** To address expert load imbalance and routing collapse, we use the same auxiliary balancing loss as in Time-MoE Shi et al. (2024), which encourages balanced expert utilization and is defined as:

$$\mathcal{L}_{\mathrm{aux}} = N \sum_{i=1}^{N} f_i P_i, \quad f_i = \frac{1}{KT} \sum_{t=1}^{T} \mathbb{I}, \quad P_i = \frac{1}{T} \sum_{t=1}^{T} s_{i,t}, \tag{12}$$

where $f_i$ represents the fraction of tokens dispatched to expert $i$, and $P_i$ is the corresponding routing probability.

Finally, the combined loss is as follows:

$$\mathcal{L} = \frac{1}{T_y} \sum_{t=1}^{T_y} \mathcal{L}_{\mathrm{ar}}(\mathbf{Y}_{revin,t}, \hat{\mathbf{Y}}_t) + \alpha \mathcal{L}_{\mathrm{aux}}, \tag{13}$$

where $\alpha$ is a hyperparameter that balances the contributions of temporal learning and expert-level regularization.

## 4 EXPERIMENTS

The experimental section of this paper conducts comprehensive evaluations across six public datasets and one self-collected dataset. Through the experiments, we would like to answer the following questions: (1) Can MoFE-Time demonstrate robust zero-shot generalization capabilities across diverse downstream tasks? (2) Can MoFE-Time achieve better performance after fine-tuned on specific tasks? (3) How do the individual contributions of each module affect the model's final performance? (4) What frequency-domain feature patterns are learned through the Frequency-Time Cell module?

### 4.1 EXPERIMENTAL SETUP

#### 4.1.1 DATASETS.

We selected Time-300B Shi et al. (2024), currently the largest and highest quality publicly available time series dataset for pre-training. We primarily utilized six public datasets and one proprietary dataset to evaluate the zero-shot and fine-tune performance of the MoFE-Time. For the public datasets, we choose ETTh1&ETTm1 Zhou et al. (2021), Weather Angryk et al. (2020), Exchange rate Lai et al. (2017), Traffic and Electricity. For the proprietary dataset, we present **NEV-sales**. We collected and organized daily store traffic data from the sales centers of a new energy vehicle company to construct the NEV-sales (New Energy Vehicle sales) dataset. Store traffic flow is a typical source of time series data with seasonal and cyclical patterns. We

| Models
Metrics | | MoFE(20M)
MSE MAE | | MoFE(100M)
MSE MAE | | MoFE(200M)
MSE MAE | | Time-MoE
MSE MAE | | Chronos-small
MSE MAE | | Chronos-base
MSE MAE | | Moirai-small
MSE MAE | | Moirai-base
MSE MAE | | TimesFM
MSE MAE | |
|---|---|---|---|---|---|---|---|---|---|---|---|---|---|---|---|---|---|---|---|
| **ETTh1** | 12 | 0.349 | 0.354 | 0.338 | 0.356 | 0.343 | 0.354 | **0.332** | **0.352** | 0.476 | 0.387 | 0.434 | 0.373 | 0.674 | 0.487 | 0.595 | 0.411 | 0.390 | 0.361 |
| | 24 | 0.320 | **0.345** | 0.315 | 0.346 | 0.322 | 0.350 | 0.320 | 0.352 | 0.436 | 0.375 | 0.399 | 0.361 | 0.557 | 0.461 | 0.405 | 0.390 | 0.361 | 0.358 |
| | 48 | 0.357 | 0.369 | 0.350 | **0.365** | **0.341** | **0.365** | 0.346 | 0.371 | 0.487 | 0.397 | 0.489 | 0.394 | 0.543 | 0.463 | 0.426 | 0.400 | 0.406 | 0.383 |
| | 96 | 0.372 | 0.383 | 0.355 | 0.378 | **0.349** | **0.376** | 0.361 | 0.386 | 0.466 | 0.409 | 0.440 | 0.393 | 0.401 | 0.402 | 0.508 | 0.408 | 0.414 | 0.404 |
| | 192 | 0.401 | 0.405 | 0.383 | 0.400 | **0.378** | **0.397** | 0.396 | 0.411 | 0.530 | 0.450 | 0.492 | 0.426 | 0.435 | 0.421 | 0.435 | 0.428 | 0.465 | 0.434 |
| | 336 | 0.424 | 0.423 | 0.402 | 0.409 | 0.399 | 0.412 | 0.412 | 0.423 | 0.570 | 0.486 | 0.550 | 0.462 | 0.438 | 0.434 | 0.457 | 0.442 | 0.503 | 0.456 |
| | 720 | 0.428 | 0.436 | 0.403 | 0.420 | 0.394 | 0.421 | 0.429 | 0.446 | 0.615 | 0.543 | 0.882 | 0.591 | 0.439 | 0.454 | 0.479 | 0.471 | 0.511 | 0.481 |
| | **AVG** | 0.379 | 0.388 | 0.364 | **0.382** | **0.361** | **0.382** | 0.371 | 0.392 | 0.511 | 0.435 | 0.527 | 0.429 | 0.498 | 0.446 | 0.472 | 0.421 | 0.436 | 0.411 |
| **ETTm1** | 12 | 0.309 | 0.323 | **0.293** | 0.322 | 0.301 | 0.322 | 0.294 | **0.320** | 0.334 | 0.334 | 0.323 | 0.333 | 0.388 | 0.370 | 0.384 | 0.371 | 0.368 | 0.346 |
| | 24 | 0.487 | 0.402 | 0.488 | 0.401 | 0.464 | 0.395 | 0.430 | 0.394 | 0.508 | 0.394 | 0.487 | 0.394 | 0.580 | 0.441 | 0.670 | 0.403 | **0.321** | **0.341** |
| | 48 | 0.465 | 0.400 | 0.493 | 0.401 | 0.438 | 0.395 | 0.398 | 0.390 | 0.534 | 0.414 | 0.469 | 0.391 | 0.673 | 0.493 | 0.739 | 0.448 | **0.346** | **0.358** |
| | 96 | 0.465 | 0.416 | 0.414 | 0.389 | 0.411 | 0.391 | 0.398 | 0.395 | 0.511 | 0.423 | 0.454 | 0.408 | 0.418 | 0.392 | 0.363 | 0.356 | **0.361** | **0.370** |
| | 192 | 0.460 | 0.423 | 0.421 | 0.409 | 0.437 | 0.414 | 0.406 | 0.410 | 0.618 | 0.485 | 0.567 | 0.477 | 0.431 | **0.405** | **0.388** | 0.375 | 0.414 | 0.405 |
| | 336 | 0.502 | 0.458 | 0.415 | 0.432 | 0.462 | 0.437 | 0.485 | 0.461 | 0.683 | 0.524 | 0.662 | 0.525 | 0.433 | 0.412 | 0.416 | **0.392** | 0.445 | 0.429 |
| | 720 | 0.604 | 0.525 | 0.527 | 0.512 | 0.621 | 0.541 | 0.691 | 0.596 | 0.748 | 0.566 | 0.900 | 0.591 | 0.462 | 0.432 | **0.460** | **0.418** | 0.512 | 0.471 |
| | **AVG** | 0.470 | 0.421 | 0.436 | 0.409 | 0.448 | 0.413 | 0.443 | 0.425 | 0.562 | 0.449 | 0.552 | 0.446 | 0.484 | 0.421 | 0.489 | 0.395 | **0.395** | **0.389** |
| **Weather** | 12 | **0.080** | 0.098 | 0.083 | 0.112 | **0.080** | 0.110 | 0.082 | 0.110 | 0.094 | **0.091** | 0.098 | 0.092 | 0.172 | 0.137 | 0.116 | 0.125 | - | - |
| | 24 | **0.110** | 0.136 | 0.133 | 0.170 | 0.115 | 0.156 | 0.125 | 0.162 | 0.128 | **0.128** | 0.136 | 0.133 | 0.141 | 0.167 | 0.139 | 0.162 | - | - |
| | 48 | **0.142** | **0.176** | 0.161 | 0.205 | 0.143 | 0.190 | 0.146 | 0.197 | 0.170 | 0.178 | 0.165 | 0.180 | 0.211 | 0.238 | 0.202 | 0.220 | - | - |
| | 96 | **0.174** | 0.220 | 0.193 | 0.252 | 0.177 | 0.237 | 0.186 | 0.245 | 0.211 | 0.243 | 0.203 | 0.238 | 0.198 | 0.222 | 0.220 | **0.217** | - | - |
| | 192 | **0.225** | 0.271 | 0.237 | 0.294 | 0.236 | 0.294 | 0.247 | 0.299 | 0.263 | 0.294 | 0.256 | 0.290 | 0.247 | 0.265 | 0.271 | **0.259** | - | - |
| | 336 | 0.294 | 0.328 | 0.311 | 0.354 | 0.317 | 0.353 | 0.322 | 0.354 | 0.321 | 0.339 | 0.314 | 0.336 | **0.283** | 0.303 | 0.286 | 0.297 | - | - |
| | 720 | 0.488 | 0.457 | 0.525 | 0.494 | 0.546 | 0.488 | 0.775 | 0.693 | 0.404 | 0.397 | 0.397 | 0.396 | 0.373 | 0.354 | **0.373** | **0.354** | - | - |
| | **AVG** | **0.216** | 0.241 | 0.235 | 0.269 | 0.231 | 0.261 | 0.269 | 0.294 | 0.227 | 0.239 | 0.224 | 0.238 | 0.232 | 0.241 | 0.229 | **0.233** | | |
| **Exchange** | 12 | 0.026 | 0.104 | 0.031 | 0.118 | 0.025 | 0.108 | 0.023 | 0.099 | 0.015 | 0.078 | **0.015** | 0.079 | 0.032 | 0.090 | 0.029 | 0.078 | 0.016 | 0.081 |
| | 24 | 0.055 | 0.150 | 0.055 | 0.162 | 0.056 | 0.159 | 0.031 | 0.122 | **0.026** | 0.107 | 0.027 | 0.110 | 0.034 | 0.124 | 0.044 | **0.098** | 0.027 | 0.109 |
| | 48 | 0.093 | 0.202 | 0.074 | 0.193 | 0.071 | 0.185 | 0.052 | 0.159 | **0.047** | 0.150 | 0.049 | 0.152 | 0.063 | 0.164 | 0.071 | **0.133** | 0.056 | 0.162 |
| | 96 | 0.167 | 0.281 | 0.137 | 0.269 | 0.139 | 0.263 | 0.123 | 0.249 | **0.091** | 0.211 | 0.093 | 0.213 | 0.109 | 0.217 | 0.121 | **0.211** | 0.110 | 0.233 |
| | 192 | 0.335 | 0.413 | 0.246 | 0.367 | 0.309 | 0.396 | 0.225 | 0.345 | **0.187** | 0.307 | 0.188 | 0.309 | 0.207 | 0.322 | 0.197 | 0.308 | 0.219 | 0.334 |
| | 336 | 0.429 | 0.504 | 0.356 | 0.436 | 0.460 | 0.512 | 0.448 | 0.494 | 0.333 | 0.416 | **0.326** | **0.412** | 0.365 | 0.428 | 0.343 | 0.426 | 0.394 | 0.456 |
| | 720 | 0.752 | 0.670 | **0.509** | 0.544 | 0.794 | 0.682 | 0.881 | 0.666 | 0.855 | 0.692 | 0.833 | 0.680 | 1.177 | 0.809 | 0.819 | 0.657 | 1.010 | 0.759 |
| | **AVG** | 0.265 | 0.332 | **0.201** | 0.298 | 0.265 | 0.329 | 0.255 | 0.305 | 0.222 | 0.280 | 0.219 | 0.279 | 0.284 | 0.308 | 0.232 | **0.273** | 0.261 | 0.305 |
| **Traffic** | 12 | 0.667 | 0.344 | **0.624** | **0.324** | 0.667 | 0.348 | 0.650 | 0.339 | 1.024 | 0.427 | 0.944 | 0.375 | - | - | - | - | - | - |
| | 24 | 0.587 | 0.305 | **0.548** | **0.279** | 0.573 | 0.302 | 0.564 | 0.293 | 0.841 | 0.362 | 0.733 | 0.296 | - | - | - | - | - | - |
| | 48 | 0.554 | 0.274 | **0.520** | 0.255 | 0.533 | 0.267 | 0.534 | 0.271 | 0.638 | 0.285 | 0.681 | **0.249** | - | - | - | - | - | - |
| | 96 | 0.589 | 0.290 | **0.525** | 0.257 | 0.531 | 0.262 | 0.550 | 0.277 | 0.572 | 0.269 | 0.607 | **0.238** | - | - | - | - | - | - |
| | 192 | 0.617 | 0.305 | **0.551** | 0.268 | 0.552 | 0.271 | 0.584 | 0.292 | 0.667 | 0.299 | 0.652 | **0.252** | - | - | - | - | - | - |
| | 336 | 0.653 | 0.315 | **0.569** | 0.274 | 0.579 | 0.283 | 0.605 | 0.299 | 0.820 | 0.344 | 0.697 | **0.265** | - | - | - | - | - | - |
| | 720 | 0.707 | 0.348 | **0.614** | 0.301 | 0.626 | 0.316 | 0.640 | 0.327 | 0.953 | 0.411 | 0.798 | **0.299** | - | - | - | - | - | - |
| | **AVG** | 0.625 | 0.312 | **0.565** | **0.280** | 0.580 | 0.293 | 0.590 | 0.293 | 0.788 | 0.343 | 0.730 | 0.282 | - | - | - | - | - | - |
| **Electricity** | 12 | 0.155 | 0.240 | **0.149** | **0.238** | 0.155 | 0.243 | 0.151 | 0.240 | 0.244 | 0.288 | 0.190 | 0.247 | 0.284 | 0.338 | 0.223 | 0.290 | - | - |
| | 24 | 0.148 | 0.232 | **0.140** | 0.228 | 0.149 | 0.236 | 0.144 | 0.233 | 0.212 | 0.265 | 0.170 | 0.232 | 0.221 | 0.309 | 0.181 | 0.267 | - | - |
| | 48 | 0.134 | 0.227 | **0.124** | 0.217 | 0.128 | 0.222 | 0.129 | 0.225 | 0.192 | 0.253 | 0.147 | 0.219 | 0.205 | 0.296 | 0.162 | 0.252 | - | - |
| | 96 | 0.141 | 0.237 | 0.133 | 0.228 | **0.132** | **0.228** | 0.136 | 0.237 | 0.311 | 0.386 | 0.160 | 0.230 | 0.205 | 0.299 | 0.158 | 0.248 | - | - |
| | 192 | 0.157 | 0.252 | 0.153 | 0.251 | **0.148** | **0.245** | 0.158 | 0.257 | 0.350 | 0.414 | 0.180 | 0.250 | 0.220 | 0.310 | 0.174 | 0.263 | - | - |
| | 336 | 0.171 | 0.249 | 0.164 | 0.262 | **0.163** | **0.261** | 0.168 | 0.270 | 0.392 | 0.445 | 0.209 | 0.275 | 0.236 | 0.323 | 0.191 | 0.278 | - | - |
| | 720 | 0.221 | 0.312 | 0.209 | 0.301 | 0.207 | 0.304 | **0.206** | 0.306 | 0.447 | 0.496 | 0.277 | 0.325 | 0.270 | 0.347 | 0.229 | 0.307 | - | - |
| | **AVG** | 0.161 | 0.253 | **0.153** | **0.246** | 0.154 | 0.249 | 0.156 | 0.253 | 0.307 | 0.364 | 0.190 | 0.254 | 0.235 | 0.317 | 0.188 | 0.272 | | |
| **NEV-sales** | 8 | 0.411 | 0.457 | 0.394 | 0.461 | 0.389 | 0.461 | **0.388** | 0.471 | 0.517 | 0.494 | 0.418 | 0.457 | 0.456 | 0.515 | 0.518 | 0.543 | 0.411 | **0.455** |
| | 16 | 0.308 | **0.406** | 0.307 | 0.416 | 0.298 | 0.412 | 0.321 | 0.439 | 0.458 | 0.471 | 0.329 | 0.418 | 0.345 | 0.460 | 0.395 | 0.482 | 0.340 | 0.424 |
| | 24 | **0.313** | **0.417** | 0.329 | 0.447 | 0.327 | 0.450 | 0.349 | 0.477 | 0.441 | 0.467 | 0.321 | 0.418 | 0.350 | 0.467 | 0.379 | 0.478 | 0.374 | 0.456 |
| | **AVG** | 0.344 | **0.427** | 0.343 | 0.441 | 0.338 | 0.441 | 0.352 | 0.462 | 0.472 | 0.477 | 0.356 | 0.431 | 0.384 | 0.480 | 0.430 | 0.501 | 0.375 | 0.445 |
| **Average** | | 0.351 | 0.339 | **0.328** | **0.332** | 0.340 | 0.338 | 0.348 | 0.347 | 0.441 | 0.369 | 0.400 | 0.337 | - | - | - | - | - | - |

Table 1: Full results of both long-term and short-term zero-shot time series forecasting experiments. The lower MSE and MAE indicate a better performance. Some results denoted by a dash (-) mean this dataset can't be evaluated because it was used by the model in pretraining. We evaluated three MoFE-Time models with different parameter scales (20M, 100M, and 200M). Detailed parameter settings are provided in the Appendix B. **Red**:the best, Blue: the 2th best.

utilized traffic data from over 400 sales centers, covering the period from their inception until December 31, 2024. During the data cleaning process, we removed store data with more than 30 consecutive days of zero values and excluded non-continuous sequences where daily data completeness was below 95%. This process resulted in the selection of 498 continuous time series at a daily granularity. The final dataset comprises a total of 330,000 time points, spanning 32 provincial-level administrative regions in China and covering major first- and second-tier cities across the country. Detailed parameters of the evaluation datasets are presented in the appendix C.

### 4.1.2 BASELINES AND EVALUATIONS

For the zero-shot experiments, we evaluate Time-MoE Shi et al. (2024), Chronos Ansari et al. (2024), Moirai Woo et al. (2024), Times-FM Das et al. (2024). For the fine-tune experiments, we evaluate Time-MoE, TimeMixer Wang et al. (2024a), TimeXer Wang et al. (2024c), TimesNet Wu et al. (2023b), and Auto-

| Models | | MoFE-time | | Time-MoE | | TimeMixer | | TimeXer | | TimesNet | | PatchTST | | AutoFormer | |
|---|---|---|---|---|---|---|---|---|---|---|---|---|---|---|---|
| Metrics | | MSE | MAE | MSE | MAE | MSE | MAE | MSE | MAE | MSE | MAE | MSE | MAE | MSE | MAE |
| ETTh1 | 12 | 0.266 | 0.324 | 0.283 | 0.350 | 0.573 | 0.499 | 0.276 | 0.335 | 0.330 | 0.381 | 0.272 | 0.332 | 0.380 | 0.426 |
| | 24 | 0.277 | 0.333 | 0.331 | 0.367 | 0.404 | 0.423 | 0.306 | 0.357 | 0.348 | 0.393 | 0.300 | 0.355 | 0.400 | 0.435 |
| | 48 | 0.311 | 0.359 | 0.320 | 0.363 | 0.359 | 0.387 | 0.354 | 0.385 | 0.379 | 0.407 | 0.347 | 0.385 | 0.433 | 0.456 |
| | 96 | 0.337 | 0.380 | 0.360 | 0.396 | 0.369 | 0.399 | 0.394 | 0.421 | 0.442 | 0.457 | 0.395 | 0.419 | 0.559 | 0.528 |
| | 192 | 0.380 | 0.410 | 0.385 | 0.412 | 0.448 | 0.429 | 0.465 | 0.475 | 0.491 | 0.489 | 0.505 | 0.495 | 0.723 | 0.646 |
| | 336 | 0.414 | 0.436 | 0.406 | 0.432 | 0.484 | 0.469 | 0.616 | 0.570 | 0.942 | 0.728 | 0.699 | 0.611 | 0.964 | 0.768 |
| | 720 | 0.453 | 0.465 | 0.456 | 0.476 | 0.498 | 0.500 | 1.034 | 0.751 | 1.194 | 0.862 | 1.111 | 0.796 | 1.402 | 0.992 |
| | AVG | 0.348 | 0.387 | 0.363 | 0.400 | 0.448 | 0.444 | 0.492 | 0.471 | 0.589 | 0.531 | 0.519 | 0.485 | 0.694 | 0.607 |
| ETTm1 | 12 | 0.213 | 0.280 | 0.217 | 0.281 | 0.231 | 0.300 | 0.180 | 0.260 | 0.197 | 0.275 | 0.229 | 0.296 | 0.389 | 0.409 |
| | 24 | 0.219 | 0.292 | 0.265 | 0.321 | 0.220 | 0.296 | 0.209 | 0.280 | 0.235 | 0.306 | 0.215 | 0.289 | 0.481 | 0.452 |
| | 48 | 0.253 | 0.324 | 0.300 | 0.343 | 0.259 | 0.322 | 0.269 | 0.328 | 0.284 | 0.344 | 0.259 | 0.321 | 0.514 | 0.496 |
| | 96 | 0.293 | 0.351 | 0.319 | 0.372 | 0.298 | 0.356 | 0.334 | 0.372 | 0.350 | 0.382 | 0.294 | 0.348 | 0.590 | 0.533 |
| | 192 | 0.332 | 0.380 | 0.358 | 0.400 | 0.359 | 0.391 | 0.374 | 0.408 | 0.519 | 0.475 | 0.392 | 0.413 | 0.693 | 0.573 |
| | 336 | 0.400 | 0.433 | 0.403 | 0.433 | 0.457 | 0.461 | 0.499 | 0.463 | 0.617 | 0.513 | 0.474 | 0.475 | 0.656 | 0.576 |
| | 720 | 0.535 | 0.514 | 0.545 | 0.500 | 0.502 | 0.482 | 0.569 | 0.508 | 0.808 | 0.622 | 0.608 | 0.549 | 0.838 | 0.678 |
| | AVG | 0.321 | 0.368 | 0.344 | 0.379 | 0.332 | 0.373 | 0.348 | 0.374 | 0.430 | 0.417 | 0.353 | 0.385 | 0.595 | 0.531 |
| Weather | 12 | 0.072 | 0.097 | 0.088 | 0.111 | 0.085 | 0.114 | 0.071 | 0.092 | 0.087 | 0.125 | 0.078 | 0.100 | 0.139 | 0.217 |
| | 24 | 0.092 | 0.124 | 0.128 | 0.158 | 0.112 | 0.156 | 0.092 | 0.124 | 0.100 | 0.145 | 0.100 | 0.137 | 0.173 | 0.260 |
| | 48 | 0.113 | 0.157 | 0.141 | 0.169 | 0.125 | 0.174 | 0.114 | 0.155 | 0.129 | 0.180 | 0.121 | 0.162 | 0.264 | 0.348 |
| | 96 | 0.148 | 0.198 | 0.178 | 0.216 | 0.146 | 0.198 | 0.152 | 0.203 | 0.158 | 0.214 | 0.156 | 0.211 | 0.362 | 0.403 |
| | 192 | 0.188 | 0.243 | 0.211 | 0.260 | 0.207 | 0.259 | 0.204 | 0.256 | 0.236 | 0.284 | 0.233 | 0.282 | 0.414 | 0.419 |
| | 336 | 0.246 | 0.287 | 0.249 | 0.297 | 0.263 | 0.307 | 0.274 | 0.315 | 0.302 | 0.333 | 0.296 | 0.328 | 0.425 | 0.438 |
| | 720 | 0.328 | 0.353 | 0.338 | 0.363 | 0.330 | 0.355 | 0.353 | 0.370 | 0.425 | 0.420 | 0.378 | 0.387 | 0.416 | 0.437 |
| | AVG | 0.170 | 0.209 | 0.190 | 0.225 | 0.181 | 0.223 | 0.180 | 0.216 | 0.205 | 0.243 | 0.194 | 0.229 | 0.313 | 0.360 |
| Exchange | 12 | 0.015 | 0.010 | 0.020 | 0.096 | 0.047 | 0.149 | 0.013 | 0.075 | 0.018 | 0.091 | 0.013 | 0.047 | 0.029 | 0.121 |
| | 24 | 0.043 | 0.141 | 0.029 | 0.119 | 0.044 | 0.153 | 0.024 | 0.105 | 0.035 | 0.134 | 0.024 | 0.104 | 0.053 | 0.169 |
| | 48 | 0.080 | 0.196 | 0.049 | 0.155 | 0.062 | 0.181 | 0.046 | 0.150 | 0.077 | 0.207 | 0.047 | 0.152 | 0.242 | 0.377 |
| | 96 | 0.082 | 0.208 | 0.094 | 0.155 | 0.114 | 0.245 | 0.106 | 0.228 | 0.220 | 0.352 | 0.097 | 0.224 | 0.997 | 0.792 |
| | 192 | 0.168 | 0.306 | 0.191 | 0.328 | 0.256 | 0.375 | 0.221 | 0.343 | 0.825 | 0.626 | 0.215 | 0.339 | 2.580 | 1.268 |
| | 336 | 0.339 | 0.429 | 0.432 | 0.490 | 0.539 | 0.557 | 0.550 | 0.511 | 1.367 | 0.824 | 0.663 | 0.610 | 1.906 | 1.093 |
| | 720 | 0.504 | 0.543 | 0.700 | 0.693 | 2.450 | 1.270 | 2.595 | 1.279 | 3.577 | 1.451 | 1.408 | 0.943 | 3.499 | 1.393 |
| | AVG | 0.176 | 0.262 | 0.216 | 0.300 | 0.502 | 0.418 | 0.508 | 0.384 | 0.874 | 0.526 | 0.352 | 0.345 | 1.329 | 0.745 |
| Traffic | 12 | 0.478 | 0.229 | 0.488 | 0.231 | 0.466 | 0.324 | 0.519 | 0.358 | 0.613 | 0.332 | 0.608 | 0.401 | 0.574 | 0.370 |
| | 24 | 0.451 | 0.215 | 0.467 | 0.218 | 0.398 | 0.287 | 0.466 | 0.332 | 0.553 | 0.309 | 0.552 | 0.406 | 0.578 | 0.377 |
| | 48 | 0.466 | 0.222 | 0.477 | 0.224 | 0.382 | 0.275 | 0.414 | 0.307 | 0.578 | 0.313 | 0.427 | 0.320 | 0.610 | 0.383 |
| | 96 | 0.486 | 0.232 | 0.504 | 0.242 | 0.383 | 0.273 | 0.374 | 0.267 | 0.603 | 0.320 | 0.368 | 0.258 | 0.648 | 0.398 |
| | 192 | 0.529 | 0.252 | 0.531 | 0.256 | 0.395 | 0.293 | 0.414 | 0.307 | 0.619 | 0.334 | 0.402 | 0.294 | 0.688 | 0.410 |
| | 336 | 0.537 | 0.263 | 0.554 | 0.267 | 0.403 | 0.292 | 0.436 | 0.324 | 0.720 | 0.372 | 0.425 | 0.312 | 0.845 | 0.471 |
| | 720 | 0.579 | 0.286 | 0.586 | 0.290 | 0.438 | 0.314 | 0.479 | 0.344 | 0.750 | 0.397 | 0.468 | 0.342 | 0.813 | 0.468 |
| | AVG | 0.504 | 0.243 | 0.515 | 0.247 | 0.409 | 0.294 | 0.443 | 0.320 | 0.634 | 0.339 | 0.464 | 0.333 | 0.679 | 0.411 |
| Electricity | 12 | 0.109 | 0.201 | 0.111 | 0.203 | 0.118 | 0.215 | 0.113 | 0.222 | 0.131 | 0.244 | 0.162 | 0.276 | 0.169 | 0.295 |
| | 24 | 0.112 | 0.202 | 0.112 | 0.203 | 0.137 | 0.234 | 0.118 | 0.232 | 0.137 | 0.248 | 0.150 | 0.255 | 0.163 | 0.281 |
| | 48 | 0.119 | 0.209 | 0.114 | 0.205 | 0.116 | 0.214 | 0.122 | 0.230 | 0.157 | 0.264 | 0.139 | 0.245 | 0.184 | 0.301 |
| | 96 | 0.129 | 0.220 | 0.137 | 0.227 | 0.159 | 0.264 | 0.141 | 0.250 | 0.190 | 0.294 | 0.136 | 0.240 | 0.208 | 0.322 |
| | 192 | 0.167 | 0.258 | 0.165 | 0.255 | 0.154 | 0.253 | 0.158 | 0.264 | 0.208 | 0.312 | 0.160 | 0.265 | 0.234 | 0.336 |
| | 336 | 0.183 | 0.277 | 0.190 | 0.283 | 0.170 | 0.269 | 0.190 | 0.297 | 0.252 | 0.342 | 0.175 | 0.278 | 0.256 | 0.357 |
| | 720 | 0.218 | 0.308 | 0.234 | 0.321 | 0.204 | 0.302 | 0.198 | 0.305 | 0.266 | 0.352 | 0.233 | 0.338 | 0.328 | 0.419 |
| | AVG | 0.127 | 0.218 | 0.152 | 0.242 | 0.137 | 0.236 | 0.130 | 0.240 | 0.165 | 0.272 | 0.149 | 0.256 | 0.220 | 0.330 |
| NEV-sales | 8 | 0.254 | 0.382 | 0.284 | 0.391 | 0.209 | 0.339 | 0.238 | 0.360 | 0.323 | 0.392 | 0.193 | 0.324 | 0.225 | 0.354 |
| | 16 | 0.187 | 0.326 | 0.223 | 0.343 | 0.268 | 0.405 | 0.321 | 0.428 | 0.413 | 0.441 | 0.367 | 0.479 | 0.296 | 0.419 |
| | 24 | 0.168 | 0.299 | 0.213 | 0.353 | 0.252 | 0.378 | 0.230 | 0.354 | 0.262 | 0.368 | 0.297 | 0.432 | 0.255 | 0.378 |
| | AVG | 0.203 | 0.336 | 0.240 | 0.362 | 0.243 | 0.374 | 0.263 | 0.381 | 0.333 | 0.400 | 0.286 | 0.412 | 0.259 | 0.384 |
| Average | | 0.264 | 0.289 | 0.289 | 0.308 | 0.322 | 0.337 | 0.338 | 0.341 | 0.461 | 0.390 | 0.331 | 0.349 | 0.584 | 0.481 |

Table 2: Finetuning results for long-term and short-term time series forecasting. Lower MSE and MAE values correspond to better performance. MoFE-Time (100M) from Table 1 is used for fine-tuning. **Red**:the best, Blue: the 2th best.

Former Wu et al. (2021). The evaluation metrics are mean square error (MSE) and mean absolute error (MAE). In our evaluation setup, prediction lengths varied among {12, 24, 48, 96, 192, 336, 720}, corresponding to context lengths of {64, 128, 256, 512, 1024, 2048, 3072} for the public datasets. Compared with existing work, we have added some shorter prediction length settings to test the robustness of the model to a wider range of prediction lengths. For NEV-sales, the context lengths and prediction are {56, 112, 180} and {8, 16, 24}. For more detailed information, please refer to the Appendix B.

## 4.2 MAIN RESULTS

### 4.2.1 ZERO-SHOT RESULTS.

We show detailed results of zero-shot prediction in Table 1. MoFE-Time achieves state-of-the-art average zero-shot inference performance across seven datasets. Notably, the MoFE-Time variant with 100M parameters demonstrates the best results, attaining average MSE and MAE values of 0.328 and 0.332, respectively which achieves average MSE and MAE reductions of 5.8% and 4.4% over the Time-MoE.

### 4.2.2 FINE-TUNE RESULTS.

We show detailed results of fine-tune prediction in Table 2. MoFE-Time also achieves state-of-the-art performance in the finetuning setting. The results show that on seven datasets, MoFE-Time attains MSE and MAE values of 0.264 and 0.289 respectively, achieves average MSE and MAE reductions of 8.6% and 6.1% over the Time-MoE. Besides, commercial forecasting of store traffic is crucial for businesses. We applied MoFE-Time to predict foot traffic in new energy vehicle (NEV) stores, leveraging its potential to enhance operational and strategic decisions. Store traffic data poses prediction challenges due to strong seasonality, long cycles, and complex noise. Despite these difficulties, MoFE-Time achieved superior performance with average MSE and MAE values of 0.203 and 0.336 which achieves average MSE and MAE reductions of 15.4% and 7.1% over the Time-MoE.

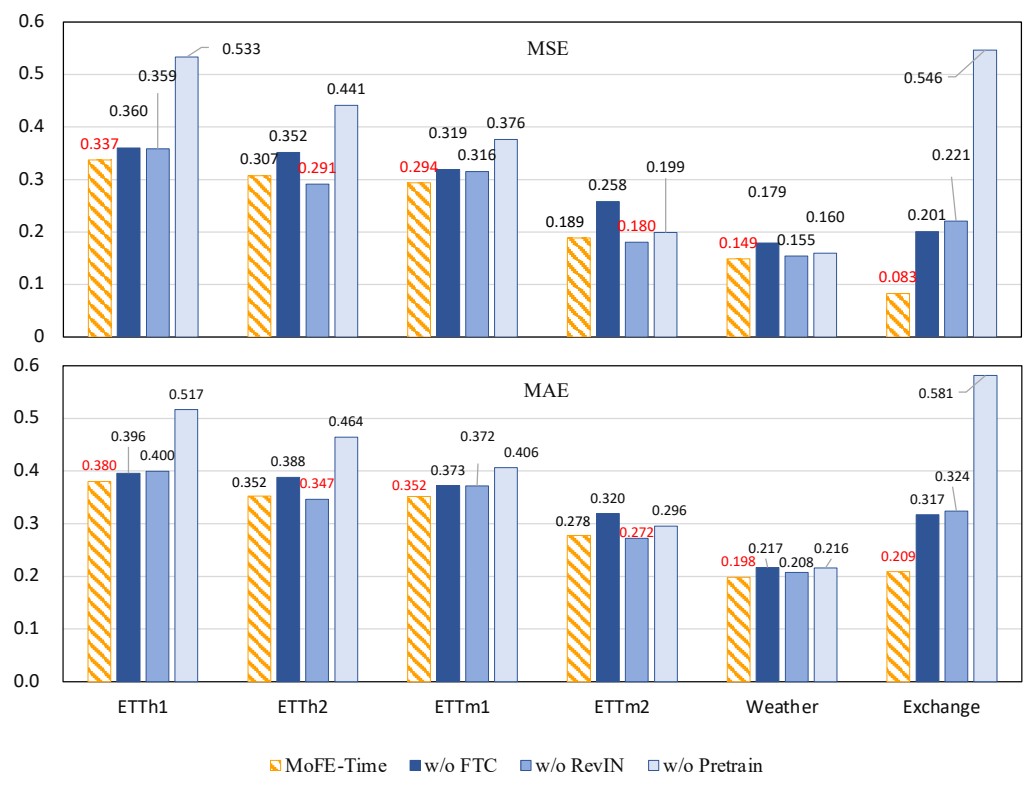

Figure 3: The ablation study of pretrain, RevIN and FTC in MoFE-Time.

## 4.3 ABLATIONS

Figure 3 show the efficacy of the pre-training, RevIN and frequency domain expert (FTC). We performed experiments under a prediction length setting of 96 with context length of 512 on six public datasets. From the result, we can conclude that most of those technical points contribute to improving the results. However, different technical points have different contributions: (1) Both metrics (MSE and MAE) on all the 6 datasets get a better result if FTC was introduced in. (2) The introduction of RevIN improved performance in 4 out of the 6 datasets. (3) The pretraining stage offers the largest contribution to both MSE and MAE of the 6 datasets. As previously mentioned, pretraining introduces external knowledge and FTC introduces a periodicity modeling

mechanism. Since both of the above two points are essential for time series prediction, they maintain consistent performance improvements across different scenarios. Since samples of Exchange-rate dataset are more unstable signals, RevIN which was designed for mitigate alleviates signal stability issues, provides the largest contribution on exchange dataset. Different from many other researchers who primarily focus on periodic modeling, our experiments demonstrate that external knowledge learning may be more important than periodicity modeling.

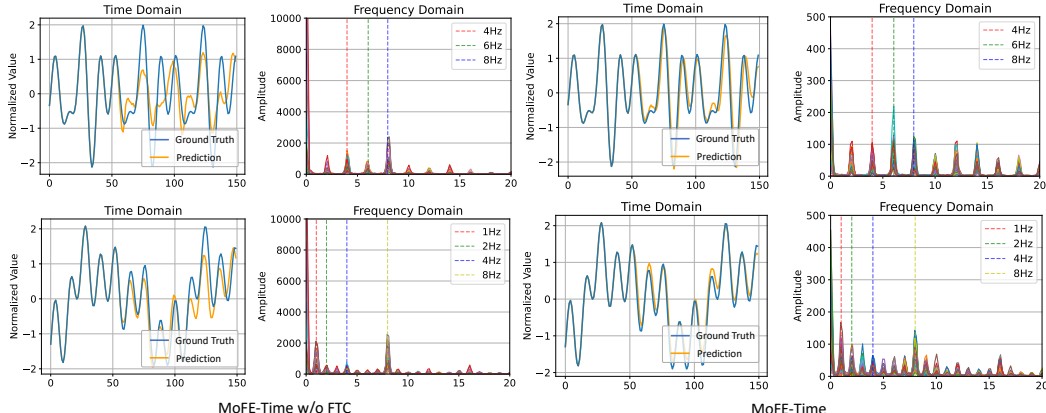

Figure 4: The visualization of frequency features between MoFE-Time and MoFE-Time w/o FTC. In the first row, we synthesized a composite sine single with 4, 6, 8 Hz, while the second row is a composite sine single with 1, 2, 4, 8 Hz. The figure illustrates the predictive results and spectral analysis of the MoFE-Time model and the MoFE-Time model w/o FTC for the two aforementioned harmonics at a context-predict length of 512-96. In the frequency domain plot, the blue dash represents the ground truth frequency distribution of the harmonic.

### 4.4 VISUALIZATION OF THE EXPERT SPECTRAL ANALYSIS

To visualize the frequency-dimensional signal characteristics, we fetched the last layer's hidden state of frequency domain experts block on MoFE-Time of two composite signals and the feedforward layer on MoFE-Time without FTC. We then applied the Fast Fourier Transform along the time dimension and compared the energy distribution across the frequency spectrum with the ground truth in Figure 4.

Two composite sinusoidal signals are synthesized at 4, 6, 8 Hz and 1, 2, 4, 8 Hz. The results show that the MoFE-Time model with the FTC module has superior prediction performance. The spectral characteristic energy of its FTC module is predominantly concentrated at the correct harmonic frequencies. In contrast, the traditional MoE architecture, which replaces the FTC module with a feedforward layer, demonstrates weaker predictive performance and less precise spectral distribution. This further confirms the effectiveness of incorporating frequency domain expert networks within the MoFE-Time model.

## 5 CONCLUSIONS

We present MoFE-Time, a novel time series forecasting model that leverages frequency domain analysis networks alongside a MoE architecture. By integrating a frequency domain analysis module within the expert networks and combining it with temporal features of signals, the model enhances its ability to learn the intrinsic properties of time series, effectively compensating for the shortcomings of existing foundation models in modeling frequency domain signals. The training architecture employs a pre-training phase followed by fine-tuning, supplemented by the integration of the RevIN method, which collectively enhances the model's predictive capabilities. MoFE-Time has achieved new state-of-the-art performance across all public benchmarks, and is especially robust on a proprietary dataset focusing on NEV sales, substantiating its effectiveness in real-world commercial applications forecasting scenarios, thus providing a valuable tool for business intelligence and operational decision-making.

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

# A  FURTHER RELATED WORK

**Time Series Forecasting**.  The current approaches to time series forecasting can be broadly categorized into three methods: (1) traditional statistical methods, (2) machine learning-based methods, and (3) deep learning-based methods.  Traditional statistical methods have advantages in both theoretical rigor and interpretability.  For example, Box et al.  introduced ARIMABox & Pierce (1970), which combines the concepts of autoregression and moving average, along with differencing operations to handle non-stationary data.  ETS (Error-Trend-Seasonality)Hyndman & Athanasopoulos (2018) is another traditional time series model that decomposes a time series into error, trend, and seasonal components.  This method is suitable for data with pronounced trends and seasonal patterns.  NPTS (Nonparametric Time Series)Härdle (1990) represents a class of nonparametric time series models that do not rely on specific functional forms or parameters.  Machine learning-based forecasting methods predict future values by fitting models to existing time series data.  For example, linear regressionMontgomery et al. (2021) is used to predict future values by building a linear regression model.  XGBoostChen & Guestrin (2016), a classical machine learning method, also performs exceptionally well in time series forecasting tasks.  Deep learning methods primarily follow the seq2seq framework to model time series forecastingSutskever et al. (2014).  In recent years, deep learning-based forecasting methods have been widely applied in time-series forecasting tasks.  Initially, recurrent neural networks (RNN) and their variants were widely employed in time series prediction tasksElman (1990); Hochreiter & Schmidhuber (1997). However, to address their limitations in long-sequence forecasting, several studies have adopted network architectures based on transformers and attention modules.  For instance, InformerZhou et al. (2021) introduced the probsparse self-attention mechanism to reduce the computational complexity of traditional self-attention, thereby addressing the memory bottleneck in long sequence tasks. AutoformerWu et al. (2021) integrated the idea of time-series decomposition into the Transformer framework and proposed the Auto-Correlation mechanism to replace traditional self-attention.

**Pre-training for Time Series Model**.  The advancement of pretrained models in natural language processing and computer vision has significantly improved the understanding of these modalities Dong et al. (2019); Selva et al. (2023).  Inspired by this progress, the field of time series forecasting has increasingly adopted self-supervised learning techniques Zhang et al. (2024).  Notable approaches include masked reconstruction Zerveas et al. (2021); Nie et al. (2023) and contrastive learning Zhang et al. (2022); Yue et al. (2022); Yang et al. (2023).  Recently, a shift towards general pretraining on extensive datasets for time series models has been observed.  Current research endeavors utilizing foundational time series models for broad-spectrum predictions:  Moment Goswami et al. (2024) stands out by employing a masked reconstruction technique, pre-trained on datasets with up to 27 billion data points, through a model architecture of 385 million parameters. Chronos Ansari et al. (2024) offers pre-trained models with up to 710 million parameters. Available in four sizes, these models effectively meet diverse application needs. Models like TimesFM Das et al. (2023) and Lag-Llama Rasul et al. (2023), which are based on decoder-only architecture, also achieved excellent pretraining performance. Time-MoE Shi et al. (2024) marks a breakthrough with its scalable architecture using a sparse mixture of experts, boosting efficiency for large-scale forecasts. Leveraging the Time-300B dataset, which includes 300 billion time points, it scales to 2.4 billion parameters, reducing inference costs significantly.

**Frequency Domain Analysis for Time Series Models**.  A series of mathematical methods centered around Fourier analysisStein & Weiss (1971); Duoandikoetxea (2024) facilitate the transformation of time series data from the time domain into the frequency domain.  As the field of temporal prediction continues to evolve, an increasing number of models are transitioning from learning time-domain signals to focusing on frequency-domain learning.  For example, Zhou et al.  proposed FEDformerZhou et al. (2022), a model that integrates a frequency domain enhancement module with transformers for improved prediction.  Wu et al.  introduced TimesNetWu et al. (2023b), which enhances the understanding and learning of time series by decomposing them into multiple periods using the Fourier Transform.  Xu et al.  applied a low-pass filter to eliminate high-frequency noise from the signal, followed by utilizing complex linear layers for prediction, in their model FITSXu et al. (2023).

However, most current methods rely on directly converting time-series signals into the frequency domain through techniques like the Fourier Transform, and then inputting these into the model for learning.  This approach does not enable the model to intrinsically learn the capability of transforming signals between the time and frequency domains.  Additionally, it poses the risk of spectral leakageOppenheim (1999); Smith et al. (1997) when learning from signals of fixed length.  To address this issue, we believe that combining a frequency-domain expert network with a mixture of experts (MoE) architecture is an effective solution.  By allowing different experts to learn the frequency domain characteristics of signals, this approach enhances the model's understanding of the intrinsic properties of the signals.

Table 3: Comprehensive Comparison of Time Series Models

| Category | Method | Architecture | Model Size (Max) | Input Token | Dataset Scale | Context Length |
|---|---|---|---|---|---|---|
| Transformer-based (Large Model) | Moirai Woo et al. (2024) | Encoder-Only | 311M | Patch | 27B/231B | 5000 |
| | Moment Goswami et al. (2024) | Encoder-Only | 385M | Patch | 1.13B | 512 |
| | Time-MoE Shi et al. (2024) | Decoder-Only | 2.4B | Point | 309B | 4096 |
| | TimesFM Das et al. (2023) | Decoder-Only | 200M | Patch | 100B | 512 |
| | Timer Liu et al. | Decoder-Only | 67M | Patch | 28B | 1440 |
| | Lag-Llama Rasul et al. (2023) | Decoder-Only | 200M | Point | 0.36B | 1024 |
| | Chronos Ansari et al. (2024) | Encoder-Decoder | 710M | Point | 84B | 512 |
| | TimeGPT Garza & Mergenthaler-Canseco (2023) | Encoder-Decoder | - | Patch | 100B | - |
| Transformer-based | PatchTST Nie et al. (2023) | Encoder-Only | - | Patch | - | - |
| | iTransformer Liu et al. (2024c) | Encoder-Only | - | Patch | - | - |
| | Autoformer Wu et al. (2021) | Encoder-Decoder | - | Patch | - | - |
| | TimeXer Wang et al. (2024c) | Encoder-Only | - | Patch | - | - |
| Non-Transformer-based | TimesNet Wu et al. (2023a) | CNN-based | - | - | - | - |
| | TimeMixer Wang et al. (2024b) | MLP-based | - | - | - | - |

# B  IMPLEMENTATION DETAILS

## B.1  TRAINING CONFIGURATION

We trained the MoFE-Time and Time-MoE under the same configuration. Both of them are trained for 200,000 steps with a batch size of 1024, and the context length is set to 4096. We apply the AdamW optimizer as optimizer with the following hyperparameters: lr = 0.001, weight_decay = 0.1, $\beta_1 = 0.9$, $\beta_2 = 0.95$. For the MoFE-Time, we present three sizes of parameters, and the detailed information is shown in Table 4.

Table 4: The detail of MoFE-Time and Time-MoE model configurations

| Model | lr | batch_size | context_length | optimizer | Layers | Heads | Experts | K | p_ratio | d_model | Activated Params | Total Params |
|---|---|---|---|---|---|---|---|---|---|---|---|---|
| Time-MoE | 0.001 | 1024 | 4096 | Adam | 12 | 12 | 8 | 2 | 0 | 384 | 50M | 113M |
| MoFE-Time(20M) | 0.001 | 1024 | 4096 | Adam | 12 | 12 | 8 | 2 | 0.06 | 384 | 12M | 22M |
| MoFE-Time(100M) | 0.001 | 1024 | 4096 | Adam | 16 | 16 | 8 | 2 | 0.06 | 768 | 48M | 118M |
| MoFE-Time(200M) | 0.001 | 1024 | 4096 | Adam | 16 | 16 | 8 | 2 | 0.06 | 1024 | 86M | 209M |

## B.2  METRICS

To evaluate the model's performance scientifically, we selected Mean Absolute Error (MAE) and Mean Squared Error (MSE) as evaluation metrics. These metrics are calculated as follows:

$$\text{MSE} = \frac{1}{H} \sum_{i=1}^{H} (x_i - \hat{x}_i)^2, \quad \text{MAE} = \frac{1}{H} \sum_{i=1}^{H} |x_i - \hat{x}_i|. \tag{14}$$

## B.3  CODE

For a fair comparison, we reproduced and evaluated the baseline results reported in all referenced papers. Specifically, for Time-MoE, as there is no publicly available pre-training code, we reimplemented its pre-training process from scratch. For other published models with open-source implementations, we utilized mature third-party repositories to reproduce their results. The links to the corresponding codebases are provided in the Table 5. To comply with the double-blind review policy, we supply anonymous links to both the third-party open-source code used for reproduction and our own implementations of Time-MoE and MoFE-Time. The MoFE-Time code will also be submitted as part of the supplementary material.

Table 5: Code source

| Source | Model | Code |
|---|---|---|
| **Ours** | MoFE-Time Time-MoE | https://anonymous.4open.science/r/MoFE-Time-6E1D |
| **Open-source** | Chronos | https://github.com/amazon-science/chronos-forecasting |
| | Moirai | https://github.com/SalesforceAIResearch/uni2ts/tree/main |
| | TimesFM | https://github.com/google-research/timesfm |
| | TimeMixer TimeXer TimesNet PatchTST AutoFormer | https://github.com/thuml/Time-Series-Library |

## C    PROCESSED DATA ARCHIVE

### C.1    PRETRAIN DATASETS

The development of robust time series foundation models mainly relies on access to massive-scale, high-integrity pre-training datasets. While the proliferation of publicly accessible time series data has increased dramatically in scale and diversity **?**, data quality presents persistent obstacles. In our paper, we used the pre-training dataset Time-300B from Time-MoE Shi et al. (2024). The Time-300B developed a streamlined data-cleaning pipeline to filter and refine raw data, and constructed the largest open-access, high-quality time series data collection for foundation model pre-training. Time-300B integrates heterogeneous real-world sources spanning critical domains including finance, web analytics, meteorology, healthcare, transportation, energy, and retail, supplemented with synthesized time series to boost both volume and representational breadth significantly. The dataset captures temporal patterns at granularities ranging from sub-second intervals to yearly aggregations, following processing via a data-cleaning pipeline, comprises more than 300 billion temporal data points, as outlined in Table 6.

Table 6: Key Statistics of the Pretraining Dataset Time-300B from Various Domains

| Domain | Energy | Finance | Healthcare | Nature | Sales | Synthetic | Transport | Web | Other | Total |
|---|---|---|---|---|---|---|---|---|---|---|
| # Sequences | 2,875,335 | 1,715 | 1,752 | 31,621,183 | 110,210 | 11,968,625 | 622,414 | 972,158 | 40,265 | 48,220,929 |
| # Observations | 15.981 B | 413.696 K | 471.040 K | 279.724 B | 26.382 M | 9.222 B | 2.130 B | 1.804 B | 20.32 M | 309.09 B |
| Proportion (%) | 5.17 % | 0.0001 % | 0.0001 % | 90.50 % | 0.008 % | 2.98 % | 0.69 % | 0.58 % | 0.006 % | 100 % |

### C.2    PUBLIC BENCHMARK DATASETS

Table 7: Key Statistics of Evaluation Datasets.

| Source | Dataset | Domain | Timesteps | Granularity |
|---|---|---|---|---|
| Zhou et al. (2021) | ETTh1 | Energy | 17,420 | 1 hour |
| Zhou et al. (2021) | ETTm1 | Energy | 69,680 | 15 min |
| Angryk et al. (2020) | Weather | Climate | 52,696 | 10 min |
| Lai et al. (2017) | Exchange | Finance | 7,588 | 1 day |
| Cuturi (2011) | Traffic | Transport | 17,544 | 1 hour |
| Trindade (2015) | Electricity | Energy | 26,304 | 1 hour |
| **Ours** | NEV_sales | Sales | 339,856 | 1 day |

We primarily utilized six public datasets and one proprietary dataset to evaluate the fine-tune performance of the MoFE-Time. For the public datasets, we choose ETTh1, ETTh2, ETTm1, ETTm2, Weather, and Exchange rate. The ETT (Electricity Transformer Temperature) series consists of an hourly dataset (ETTh1) and a 15-minute level dataset (ETTm1). Each contains load features of seven oil and electricity transformers from July 2016 to July 2018. The Weather dataset includes 21 weather indicators with data recorded every 10 minutes in 2020. The Exchange rate dataset includes daily exchange rate data of eight countries from 1990 to 2016. The Traffic describes road occupancy. The dataset contains hourly data recorded by highway sensors in San Francisco from 2015 to 2016.

#### C.2.1    NEV-SALES

To comprehensively and rigorously assess the effectiveness and generalization capability of the time series forecasting method proposed in this paper within real-world commercial scenarios, we have constructed and publicly released a novel large-scale footfall dataset, designated as NEV-sales. This dataset is sourced from the intelligent monitoring systems deployed at Li Auto Inc. sales centers across mainland China, thereby ensuring the authenticity, authority, and practical value of the data.

Data Collection and Composition: The NEV-sales dataset meticulously records the daily customer footfall of 498 independent sales centers over the period from May 2, 2024, to December 16, 2024. Data is automatically collected and aggregated by internal footfall monitoring devices installed at each sales center, effectively minimizing biases that may arise from manual counting. In total, the dataset comprises 109,238 daily footfall data points, forming 498 independent time series with direct business relevance.

Dataset Characteristics and Challenges: The NEV-sales dataset exhibits the inherent complexities of real-world time series data, thereby providing rich challenges for model evaluation:

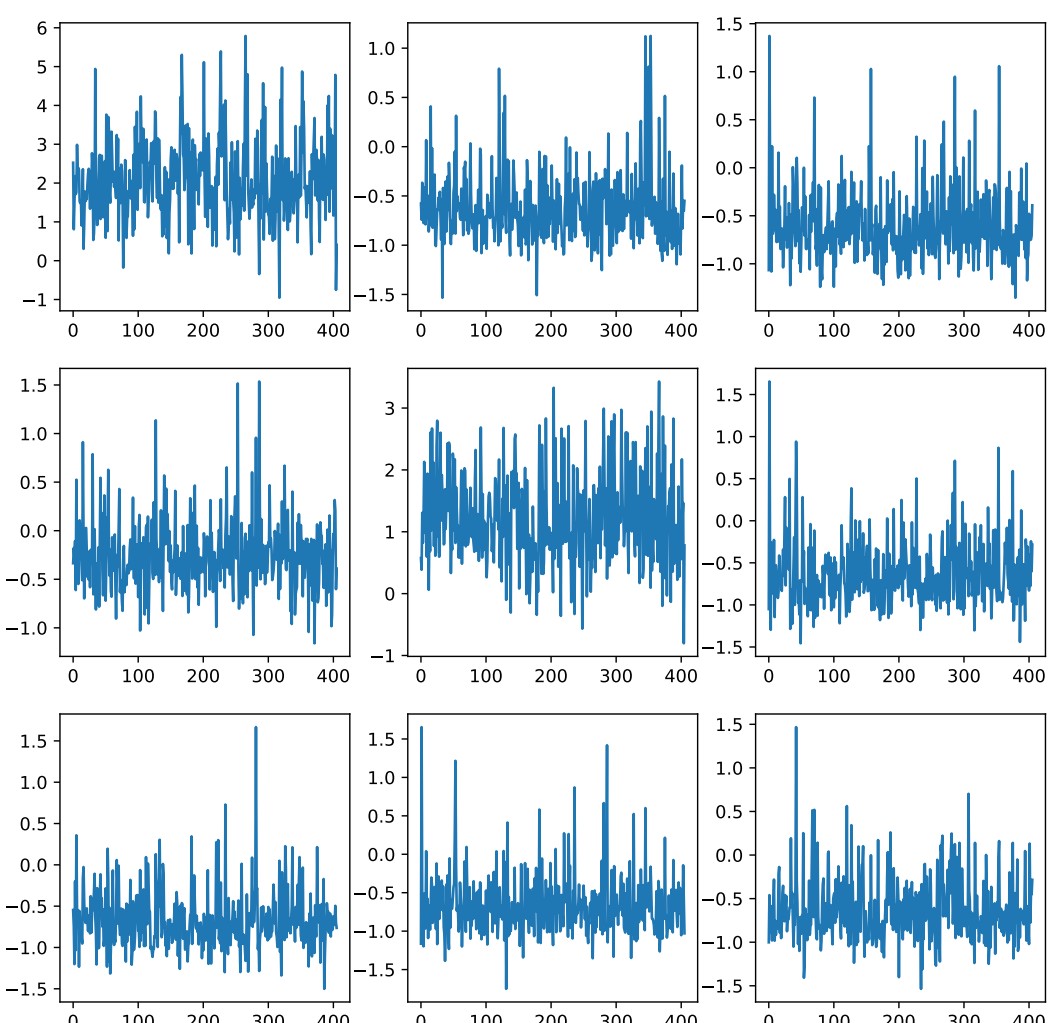

Figure 5: Visualization of representative samples from NEV-sales

Variable Length and Missing Values: Due to differences in opening dates among sales centers and occasional device maintenance, the lengths of the time series vary, ranging from 112 days to 229 days, with an average sequence length of 219.35 days. This variability and implicit missingness require models to possess robustness in handling irregular time series data. Multiple Periodicities and Trends: The footfall data is clearly influenced by multiple temporal factors, including intra-week effects with a 7-day periodicity (e.g., significant increases in visitor counts on weekends compared to weekdays), event-driven fluctuations due to public holidays, and possible long-term trends induced by marketing campaigns, seasonal variations, or new vehicle launches. Stochasticity and Noise: As authentic commercial data, the footfall figures are inevitably affected by a variety of uncontrollable factors, such as weather conditions and local unforeseen events, leading to pronounced randomness and noise. This presents higher demands on the model's resilience to disturbances and prediction stability. Large Scale and Diversity: The dataset covers nearly 500 sales centers distributed across different geographic locations, city tiers, and surrounding environments, resulting in diverse footfall patterns. This diversity provides an ideal testbed for examining the generalizability and scalability of forecasting models across heterogeneous real-world scenarios.

# D ADDITIONAL RESULTS

## D.1 LOSS ABLATION STUDY

In conventional Time-MoE model architectures, the Huber loss is typically employed as the regression loss function. The details of the Huber loss are as follows.

$$\mathcal{L}_{\mathrm{ar}}(y_t, \hat{y}_t) = \begin{cases} \frac{1}{2}(y_t - \hat{y}_t)^2, & \text{if } |y_t - \hat{y}_t| \leq \delta, \\ \delta \times (|y_t - \hat{y}_t| - \frac{1}{2} \times \delta), & \text{otherwise,} \end{cases} \quad (15)$$

where $\hat{y}_t$ denotes the actual data point at time $t$, and $y_t$ is the predicted value.

In MoFE-Time, we utilize the Cauchy loss as the regression loss function. We conducted a series of ablation experiments to compare the performance of the Cauchy loss with that of the Huber loss. In these experiments, we trained a MoFE-Time model with 100M parameters under identical training configurations. The specific results are presented in Figure X. The results indicate that, under the same experimental settings, the average performance of models trained with the Cauchy loss surpasses that of those trained with the Huber loss.

Table 8: MSE and MAE for 512-96 forecasting in ETTh1 dataset with different losses

| MoFE-Time | ETTh1 | |
|---|---|---|
| | MSE | MAE |
| Huber Loss($\delta = 2$) | 0.3648 | 0.3908 |
| Huber Loss($\delta = 0.3$) | 0.3671 | 0.3860 |
| Huber Loss($\delta = 0.2$) | 0.3679 | 0.3852 |
| Cauchy Loss($c = 2$) | **0.3523** | **0.3833** |

## D.2 INFERENCE TIMING

We utilized MoFe-Time(100M) and Time-MoE with 113.35 M parameters(As shown in Table 4). Inference time is shown in Figure 6. In these plots, we report the inference time on the ETTh1, ETTh2, and ETTm1 datasets, we time our method and Time—MoE using an A100(40GB) GPU. MoFe-Time demonstrates equal or even faster inference speeds compared to Time-MoE under the same scale of parameters.

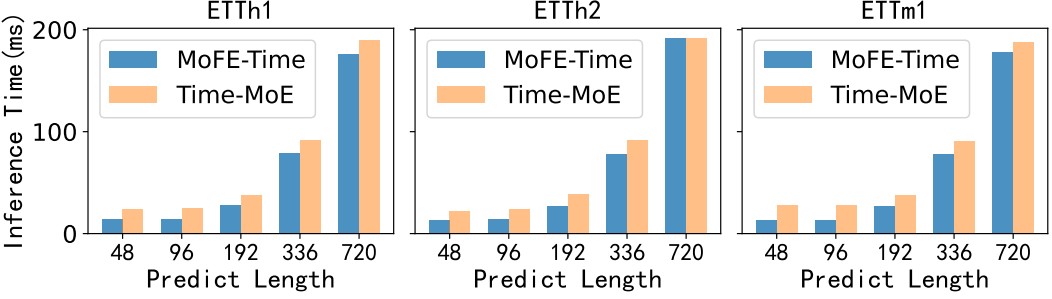

Figure 6: Comparison of Inference Speed per Data Point between MoFE-Time and TimeMoE

## D.3 FORECAST SHOWCASES

To visualize the performance differences among various large-scale time series models, we present the forecasting results of our model. Alongside MoFE-Time's results, we also show the performance of other baseline models. The Figure 7 and **??** provide a comprehensive view of their comparative capabilities.

## D.4 APPLICATIONS OF MoFE-TIME IN OTHER REAL-WORLD SCENARIOS

In addition to NEV-Sales, MoFE-Time has also been widely applied across various commercial scenarios within our organization. Notably, it achieved excellent results in addressing the parts demand forecasting problem for the PDC warehouse. We predicted the demand for PDC warehouse parts for the period from March 19 to March 25, 2025. Accurate demand forecasting enables reasonable inventory preparation, which not only

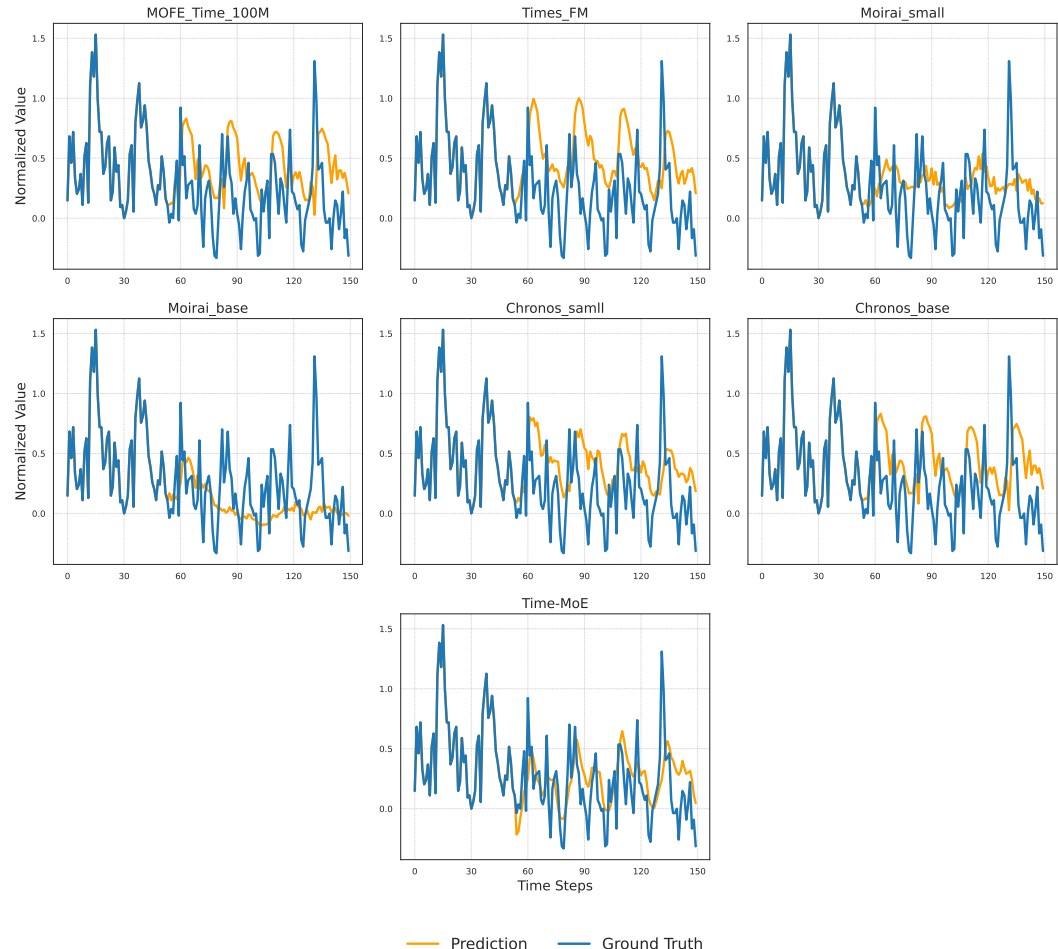

Figure 7: Zero shot time series forecasting cases from ETTh1 by different models with context-predict length = 512-96.

reduces warehousing costs but also improves service satisfaction. The results demonstrate that MoFE-Time achieves higher accuracy compared to both the online baseline models and forecasts made by domain experts. The full result is shown in Table 9. The definition of precision is as follows:

$$\text{Precision} = 1 - (\hat{x} - x)/\hat{x}, \tag{16}$$

where $\hat{x}$ represents the actual outbound quantity and $x$ represents the predicted outbound quantity.

| Component Type | Actual Outbound Quantity | Predicted outbound quantity | | | Precision | | |
|---|---|---|---|---|---|---|---|
| | | MoFE-Time | Hunman Evaluate | Baseline | MoFE-Time | Hunman Evaluate | Baseline |
| AA | 144467 | 158166.981 | 219191 | 166594 | **91%** | 48% | 85% |
| A | 10435 | 12089.116 | 13962 | 14287 | **84%** | 63% | 66 % |
| B | 3640 | 3488.379 | 3120 | 4786 | **96%** | 86% | 69% |

Table 9: Performance of MoFE-Time on Parts Demand Forecasting Tasks.

