# OpenReview forum: "MoFE-Time: Mixture of Frequency Domain Experts for Time-Series Forecasting"
_ICLR.cc/2026/Conference — ICLR 2026 Conference Withdrawn Submission_

### Official Review · Reviewer_BmbN · 2025-10-27

**Soundness:** 3
**Presentation:** 3
**Contribution:** 2
**Rating:** 4
**Confidence:** 4

**Summary:**

This paper introduces MoFE-Time, which combines temporal and frequency representations in a Mixture-of-Experts framework to improve time series forecasting performance. It presents a Frequency-Time Cell (FTC) within the MoE framework to capture frequency-domain and time-domain characteristics of signals jointly. The model is pretrained on the Time-300B dataset and evaluates across six public benchmarks. It also presents the NEV-sales dataset for evaluation. It achieves state-of-the-art (SOTA) time series forecasting performance on both zero-shot and fine-tuning settings.

**Strengths:**

1. It considers frequency domain features in MoE design, which is an incremental contribution to existing works.
2. It has a comprehensive experiment across short- and long-horizon settings and in zero-shot and fine-tuning settings. It also introduces a new dataset to demonstrate the model’s applicability in real-world scenarios.
3. Through the ablation study, it clearly shows that each component: RevIN, FTC and pretraining has positive effects for forecasting. The visualization further supports the interpretability of the frequency-domain experts.

**Weaknesses:**

1. The proposed model is heavily based on Time-MoE and the components of RevIN, and the time-to-frequency domain transformation is well-studied in time series forecasting. This FTC is only an incremental modification.

2. It's unclear how harmonics are learned or updated during training, and could explain more in section 3.2

3. Missing complexity analysis, could you include analysis such as effective FLOPs per time series token to give an idea that introducing FTC would not seriously introduce computational cost?

**Questions:**

See weakness.

---

### Official Review · Reviewer_p4mT · 2025-10-27

**Soundness:** 2
**Presentation:** 3
**Contribution:** 1
**Rating:** 2
**Confidence:** 4

**Summary:**

This paper adopts Frequency and Time Cells (FTC) into MoE to integrate time and frequency features for time series forecasting foundation model. The experiments are conducted on 7 datasets with zero-shot and finetuning settings.

**Strengths:**

1. This paper is easy to follow.
2. The evaluation settings are extensive.
3. Code and data are provided for reproducibility.

**Weaknesses:**

1. The claimed contributions in this paper say that "capture frequency-domain characteristics through Fourier transforms". However, this has been well studied in time series forecasting models, such as [1], and time series pre-training or foundation models, such as [2-3].  The claimed contributions and novelty are limited.

[1] Frequency enhanced decomposed transformer for long-term series forecasting. In ICML, 2022.

[2] Self-supervised contrastive pre-training for time series via time-frequency consistency. NeurIPS, 2022.

[3] Towards a General Time Series Forecasting Model with Unified Representation and Adaptive Transfer. In ICML, 2025.

2. Key experimental comparison for baselines [2-3] is missing. Moreover, I briefly checked the finetuning settings where the performance of this work is worse than [3].

3. This work uses the Cauchy loss, which is proposed by previous work, to get better performance. The compared baselines may be also improved with this loss.

**Questions:**

see weaknesses.

---

### Official Review · Reviewer_nxHy · 2025-10-28

**Soundness:** 3
**Presentation:** 3
**Contribution:** 2
**Rating:** 6
**Confidence:** 4

**Summary:**

The paper presents MoFE-Time, a time-series forecasting model that combines frequency-domain decomposition with a mixture-of-experts routing scheme.
The central component, named the Frequency-Time Cell (FTC), allows different experts to specialize in specific frequency bands, while a gating module adaptively fuses their outputs in the time domain.
The model also adopts Reversible Instance Normalization and temporal aggregation layers to manage scale variations and non-stationarity.

Empirical evaluations cover six standard benchmarks and one proprietary dataset (NEV-sales). MoFE-Time achieves competitive or superior performance compared to several strong baselines, including TimeMixer, Time-MoE, TimesNet, and PatchTST. Visualizations show interpretable frequency responses of different experts.

**Strengths:**

**Sound and practical motivation**.
The paper recognizes a real weakness in current foundation models for time-series — the lack of explicit spectral reasoning — and attempts a principled remedy.

**Consistent empirical gains**.
The experiments are numerous but not excessive; results are stable across horizons, which indicates robustness.

**Interpretability**.
The spectral visualizations give a genuine sense that the frequency experts are learning meaningful decompositions.

**Weaknesses:**

**No strong theoretical insight**.
The intuition behind the frequency gating is good, but the paper stops short of explaining why mixture routing is beneficial in the spectral domain.
Even a brief analysis of frequency sparsity or aliasing mitigation would strengthen it intellectually.

**Ablation depth**.
Most ablations isolate significant components (FTC, RevIN). It would be more illuminating to see fine-grained variations — e.g., what happens when experts overlap in frequency support or when gating becomes too dense.

**Scalability discussion is missing**.
Since the model scales up to 200M parameters, a small comment on compute or routing efficiency would help readers assess practical feasibility.

**Questions:**

1. Have you considered adding Time-LLM, S2IP-LLM in your evaluation, given their conceptual overlap in spectral modeling?

2. Does the model’s advantage hold in very long-horizon forecasts (say, 1k–2k steps)?

3. What is the impact of increasing or reducing the number of frequency experts — is the model sensitive to over-partitioning?

4. How stable is the MoE routing when fine-tuning on small datasets (like NEV-sales)?

5. Can the authors elaborate on whether Fourier decomposition is applied globally or adaptively across segments?

---

### Official Review · Reviewer_t9WB · 2025-10-31

**Soundness:** 2
**Presentation:** 2
**Contribution:** 2
**Rating:** 2
**Confidence:** 4

**Summary:**

The authors build a Time series Foundation Model (TSFM) which is based on the Time-MoE architecture and the contribution of the authors, the Frequency and Time Cell (FTC). The resulting model is then called MoFE-Time, to reflect the basis Time-MoE and the FTC. The central point of the FTC is to decompose a token of the time series into its frequency components and then pass both, the time and the frequency components through linear layers and combine it finally. With this approach, the authors aim to "to enhance the model’s capacity for representation and learning".

**Strengths:**

The works builds upon the shoulders of giants. The authors reuse the Time-MoE architecture and also their pretraining datasets and introduce a new set of experts. Therefore, the performance on the individual datasets that they tested on, appears to be reasonably good.

**Weaknesses:**

However, the strength mentioned above, is immediately the strongest weakness of the paper also, because the novelty of this work is very limited. The only real contribution of the authors is to replace the experts from the Time-MoE with something else. Moreover, the experts in the original Time-MoE architecture where linear layers and now the authors added the frequency decomposition to it. Other works have also looked in to the decomposition of a timeseries into its frequency components and then used this for more performant models, i.e., FEDFormer, which the authors also mention. However, the description of the FTC in section 3.2 feels like a standard description from the time domain into the frequency domain and from this description alone it remains unclear why the approach of the authors should work better in any way. Furthermore, the authors mention the separate decomposition using the sine and cosine harmonics used in the FTC only in the very last sentence and do not elaborate further why this may be important. Thus, the description in the text and the illustration in Figure 2 show raise some questions.

It is a pity that although the authors worked with a relatively large and powerful foundation model, they only evaluated its performance on small individual datasets and not on the common real-world benchmarks for timeseries, such as the GIFT-Eval. From these limited results it is hard to judge the real capabilities of this model. Furthermore, the result tables are inconsistent and raise a series of questions, see below.

**Questions:**

Regarding the FTC:\
Could you please elaborate what makes the formulation in Section 3.2 special?\
Why was a separate decomposition with sine and cosine harmonics chosen and what is its performance impact?

Regarding evaluations:\
Why has the foundation model not been evaluated on common benchmarks for timeseries foundation models?\
Why are several stronger baselines, such as the iTransformer [1], TTM [2] and others not part of the comparison?\
Please clarify why the performance number of the central Time-MoE architecture are not identical to the original paper (in most cases of Table 1 they are worse or different than in the original paper in Table 3 for example).\
Please correct the wrong coloring scheme, indicating the best and second best architecture. In several instances this is incorrect and misleading. For example, Table 2, for the Exchange dataset in the line with 48, TimeXer should be: MSE 1) TimeXer 2) PatchTST and there are other places as well.\
Please clarify why the performance of the fine-tuned models are worse than the base model, compare Table 2 with Table 3. For example, the performance of the Time-MoE baseline is often worse in Table 2 than in Table 1. Electricity 720 Time-MoE: Table 2: 0.234; 0.321 while in Table 1: 0.206; 0.306.

[1] Liu, Yong, et al. "iTransformer: Inverted Transformers Are Effective for Time Series Forecasting." arXiv, 10 Oct. 2023, doi:10.48550/arXiv.2310.06625.
[2] Ekambaram, Vijay, et al. "Tiny Time Mixers (TTMs): Fast Pre-trained Models for Enhanced Zero/Few-Shot Forecasting of Multivariate Time Series." arXiv, 8 Jan. 2024, doi:10.48550/arXiv.2401.03955.

---

### Note · Authors · 2026-01-04

I have read and agree with the venue's withdrawal policy on behalf of myself and my co-authors.